# Differentially Expressed Candidate miRNAs of Day 16 Bovine Embryos on the Regulation of Pregnancy Establishment in Dairy Cows

**DOI:** 10.3390/ani13193052

**Published:** 2023-09-28

**Authors:** Vanmathy R. Kasimanickam, Ramanathan K. Kasimanickam

**Affiliations:** 1Center for Reproductive Biology, College of Veterinary Medicine, Washington State University, Pullman, WA 99164, USA; vkasiman@wsu.edu; 2AARVEE Animal Biotech LLC, Corvallis, OR 97333, USA; 3Department of Veterinary Clinical Sciences, College of Veterinary Medicine, Washington State University, Pullman, WA 99164, USA

**Keywords:** dairy cows, fertility, micro-RNA, mRNA, bioinformatics, embryo, placenta, fetal organs, development

## Abstract

**Simple Summary:**

MicroRNAs (miRNAs) are small, noncoding RNAs and their role in embryo implantation is unknown. The study objective was to identify differentially expressed (DE) miRNAs between day 16 competent, elongated embryos from normal cows and day 16 noncompetent, tubular embryos from repeat breeder cows, integrate DE-miRNAs to their target genes, and categorize the target genes based on predicted biological functions. Out of 84 bovine-specific, prioritized miRNAs analyzed by RT-PCR, 19 were differentially expressed in the competent embryos compared to noncompetent ones (*p* ≤ 0.05; fold regulation ≥ 2 magnitudes). Top-ranked integrated genes of DE-miRNAs predicted to regulate various biological and molecular functions, cellular processes, and pathways. The categorized groups of genes may be associated with crucial signaling pathways regulating the development of embryo, placenta, and various organs. In conclusion, DE-miRNAs in day 16 bovine embryos may control pregnancy establishment, interacting with genes and transcription factors.

**Abstract:**

Recent advances in high-throughput in silico techniques translate experimental data into meaningful biological networks through which the role of individual proteins, interactions, and their biological functions are comprehended. The study objective was to identify differentially expressed (DE) miRNAs between the day 16 competent, elongated embryo from normal cows and the day 16 noncompetent, tubular embryos from repeat breeder cows, assimilate DE-miRNAs to their target genes, and group target genes based on biological function using in silico methods. The 84 prioritized bovine-specific miRNAs were investigated by RT-PCR, and the results showed that 19 were differentially expressed (11 up- and 8 down-regulated) in the competent embryos compared to noncompetent ones (*p* ≤ 0.05; fold regulation ≥ 2 magnitudes). Top-ranked integrated genes of DE-miRNAs predicted various biological and molecular functions, cellular processes, and signaling pathways. Further, analysis of the categorized groups of genes showed association with signaling pathways, turning on or off key genes and transcription factors regulating the development of embryo, placenta, and various organs. In conclusion, highly DE-miRNAs in day 16 bovine conceptus regulated the embryogenesis and pregnancy establishment. The elucidated miRNA-mRNA interactions in this study were mostly based on predictions from public databases. Therefore, the causal regulations of these interactions and mechanisms require further functional characterization.

## 1. Introduction

Establishment of pregnancy includes pregnancy recognition, implantation, and placentation. Embryonic cells proliferate and differentiate to form fetus and placenta during early embryogenesis. Early embryogenesis, from fertilization to implantation, comprises diverse morphological, cellular, genomic, and biochemical changes [1,2,3]. These changes contribute to the elongation of embryo, cell-to-cell interaction between the uterus and the embryo, and development of placenta for ascertaining pregnancy. The bovine embryos at the blastocyst stage on day 7 are 150 to 200 μm in diameter but the whole conceptus becomes approximately 6 cm long around day 16 [4,5]. Conceptus elongation involves exponential increases in length and weight of the trophectoderm and onset of extraembryonic membrane differentiation, including gastrulation of the embryo and formation of the yolk sac and allantois that are vital for embryonic survival and formation of a functional placenta.

Reprogramming of the genome with the organized temporal and spatial gene expression changes is crucial for embryonic development [6,7,8,9]. Vast numbers of specific genes are expressed and those genes change their expression levels specific to embryonic stages to support the complex mechanisms of embryogenesis and implantation. Further, the differentiation of trophoblastic cells and the formation of placenta are tightly controlled by precisely turning on or off appropriate sets of genes during embryogenesis. Dysregulated expression of genes may result in early embryo loss [10,11,12,13]. Aberrant gene expression patterns were associated with placental abnormalities [14]. Although numerous molecules participate in trophoblast differentiation and placentation, the precise molecular and genetic pathways which lead to the implantation and formation of placenta remains elusive. Embryonic mortality is one of the biggest limiting factors in dairy sector profitability [15]. A detailed investigation of molecular mechanisms involved in the regulation of pre-implantation embryo is vital to furthering the knowledge of the embryogenesis and implantation processes. The findings from such investigation may assist in comprehending the reprogramming of embryogenesis and can serve as a basis for more hypothesis-driven research to elucidate the molecular biology of embryogenesis.

Hence, the goal of this study was to associate differentially expressed bovine-specific miRNAs between day 16 embryo of fertile and repeat breeder cows with the crucial functions of embryogenesis and embryonic development. Further, we aimed to perform qPCR and Western blot on selected candidates of genes and proteins to substantiate their functionalities.

## 2. Materials and Methods

The study protocol was approved by the institutional animal care and use committee of Washington State University (04479).

### 2.1. Cow Selection, Breeding and Embryo Collection

Cow selection, breeding, and embryo collection methods were similar to those described previously [8]. Briefly, based on on-farm performance records, repeat breeder (sub-fertile; with a history of failing to maintain a pregnancy following three consecutive inseminations and a documented embryonic loss between 30 and 60 days [10,16]) and normal Holstein dairy cows (fertile; with a history of conceiving with the first or second insemination and without documented embryonic loss) were selected. At the time of enrolment, cows were in their third lactations, between 60 and 100 days in milk and were housed in free-stall barns. Cows included in the study were free of peripartum metabolic disorders, dystocia, retained placenta, early postpartum uterine diseases, mastitis, or lameness during previous pregnancies and lactations. Additionally, these cows were apparently healthy and their body condition score ranged from 2.5 to 3.5 (5-point score: 1 emaciated to 5 obese) [17,18]. All of the cows were fed twice daily with a TMR formulated to meet or exceed dietary needs for cows weighing ~520 to 710 kg and the average milk production was 32.3 ± 2.6 and 32.3 ± 2.2 kg of 3.5% fat-corrected milk for normal and repeat breeder groups, respectively.

As per routine breeding management of the farm, the selected cows were artificially inseminated following implementation of the Select Synch + CIDR protocol [10,19]. A single AI sire (sire conception rate score ≥+3) was used to inseminate the cows. The cows were inseminated 12 h after the first sign of standing estrus. It should be noted that ovulation occurs approximately 30 h after standing estrus can first be observed [20,21]. The embryos were recovered using the non-surgical uterine flush technique on day 16 after insemination. Briefly, conceptuses were collected by standard non-surgical uterine flushing technique using an 18 g catheter (Ag Tech Inc., Manhattan, KS, USA) in phosphate-buffered saline (PBS; pH 7.4) [10,22,23]. Care was taken while flushing uteri to ensure the recovery of intact conceptuses, especially by controlling the flow of flush medium entering the uterus and flow of recovery. The length of the conceptuses was measured and they were categorized as tubular (10 to 20 mm) or filamentous (≥25 mm) embryos [8,24]. A subset of day 16 embryos collected for our previous study [10] were also used for miRNA analysis for this study. The embryo samples were snap-frozen and stored at −80 °C until further use. At the time of analysis, each embryo was thawed at room temperature and sonicated by exposure to sound wave bursts for 6 s until visibly disrupted. Each embryo sample was aliquoted for miRNA, mRNA, and protein analysis.

### 2.2. Mature Bovine miRNA Expression Profile

Day 16 tubular embryos (n = 4) from subfertile, repeat breeder cows and filamentous embryos (n = 4) from fertile, normal cows were analyzed singly for the determination of miRNA. For detailed RT-PCR methods, refer to previous reports [10,25,26]. Briefly, total RNA including small RNAs were isolated using a miRNeasy kit (Qiagen, Valencia, CA, USA). The RNA was reverse transcribed using miScript II RT kit (Qiagen, Valencia, CA, USA). A HiSpec buffer (5×) was used to prepare cDNA for mature miRNA profiling. miRNAs in day 16 conceptus were identified using the MiScript miRNA PCR array method. It should be noted that this expression profiling consisted of real-time PCR, so single miRNA validation experiments were not necessary. A bovine miRBase profiler plate (Appendix A, Qiagen, Valencia, CA, USA) consisting of specific primers for 84 highly prioritized bovine mature miRNAs was selected from the most current miRNA genome, as annotated in miRBase V.20.0 & V.22.1 (www.miRBase.org). The following small RNAs were included as control: cel-miR-39-3p, Small Nucleolar RNA C/D box (SNORD) 42B, SNORD69, SNORD61, SNORD68, SNORD96A, RNA U6 Small Nuclear 2 (RNU6-2), microRNA reverse transcription control (miRTC), and positive PCR controls (PPC).

#### Data Analysis

Raw CT in .XLS file format was uploaded to the data analysis center. Data quality control was examined, and the CT values of samples were calibrated to the CT values of cel-miR-39-3p. Global CT mean of expressed miRNAs, was chosen to normalize the target miRNAs. To identify DE miRNAs (up- and downregulated) between elongated and tubular embryo groups, the distribution of CT values and raw data means were reviewed. Average ΔCT, 2^−ΔCT^, fold change (≥2), and *p*-value (≤0.05; Students *t*-test) were calculated. Statistical analysis was performed using SAS Analytics software (9.4 version; SAS Institute, Cary, NC, USA). To detect ≥2-fold mean differences (SD ± 0.4) in miRNA expressions differences with adequate statistical power (1 − β = 0.8) and statistical significance (α = 0.05), a minimum of four samples per group were needed [27,28].

### 2.3. Bioinformatics Analysis

#### 2.3.1. Conserved Nucleotide Sequences

Nucleotide sequences of bovine encoded DE miRNAs were retrieved from miRBase, (www.mirbase.org; accessed on 10 July 2023) and compared with those of *homo sapiens* for similarities [29,30].

#### 2.3.2. Prediction and Analysis of Target Genes of DE miRNAs

The miRNA–gene interaction analysis was performed for DE miRNAs using miRNet (http://www.mirnet.ca/ accessed on 10 July 2023) [31]. This tool consisted of information from different miR databases, such as TarBase, miRTarBase, and miRecords. The degree and betweenness of target genes were also determined. The target genes from the analysis were used for further determining protein–protein interaction (PPI).

#### 2.3.3. Construction of the Protein–Protein Interaction Network and Screening of Hub Genes

The online database search tool for the retrieval of interacting genes/proteins (STRING) (http://stringdb.org/; accessed on 11 July 2023) was used to elucidate the PPI of target genes [32]. Protein–protein functional association (extended version of PPI) was performed by using gene ontology (GO terms) functional annotation for biological process and the Kyoto Encyclopedia of Genes and Genomes (KEGG) pathway enrichment. *p* < 0.05 was regarded as statistically significant.

The PPI STRING network was exported to the Cytoscape (version 3.10; https://cytoscape.org/; accessed on 11 July 2023) visualizing platform [33]. The hub genes were selected out as the top 20 nodes of the PPI network using the maximal clique centrality (MCC) method (cytoHubba; https://apps.cytoscape.org/apps/cytohubba; accessed on 11 July 2023), which performed better on the precision of predicting top essential proteins [34]. Maximal clique is the complete subgraph of a given graph which contains the maximum number of nodes.

### 2.4. Real-Time Polymerase Chain Reaction for Determining mRNA Expression of Target Genes

Target genes such as the Peroxisome proliferator-activated receptor (*PPAR*), *A, B,* and *G*; RXR—Retinoid X receptor (*RXR*) *A* and B; Radical S-Adenosyl Methionine Domain Containing 2 (*RSAD2*); Solute Carrier Family 2 Member 1 (*SLC2A1*); Solute Carrier Family 27 Member 6 (*SLC27A6*); C-X-C motif chemokine ligand 10 (*CXCL10*); Interferon interferon-stimulated gene-15 (*ISG15*), DNA methyltransferase 1 (*DNMT1*); Zinc Finger E-Box Binding Homeobox 1 (*ZEB1*); and Hypoxia Inducing factor 1A (*HIF1A*) predicted by DE-miRNAs were selected from the group of target genes to substantiate the mRNA expressions in day 16 embryos.

Total RNA extraction and complementary DNA synthesis were performed as previously described [10,26]. Briefly, aliquots of the elongated and tubular day 16 embryo samples were used to extract RNA by TRizol (Invitrogen, Carlsbad, CA, USA). The RNA concentration and quality were determined using NanoDrop-1000 spectrophotometer (Thermo Scientific Inc., Waltham, MA, USA) and all RNA samples were treated with DNAse I (Invitrogen) to remove the DNA contaminant. Complementary DNA was synthesized using the iScript cDNA synthesis kit (Bio-Rad Laboratories Inc., Hercules, CA, USA) and stored at −20 °C.

Primer-BLAST (www.ncbi.nlm.nih.gov/tools/primer-blast/, accessed on 1 July 2022) was used to design specific primer pairs for the target genes (Appendix A). Appendix A shows the ethidium bromide-stained polyacrylamide gel-revealing band at the expected size of amplification product for each of the GAPDH and the target DNAs. Real-time PCR was carried out using Fast SYBR Green Master Mix (Applied Biosystems, Foster City, CA, USA) as previously described [8], following the manufacturer’s instructions. Endogenous control glyceraldehyde-3-phosphate dehydrogenase (GADPH) was used to normalize the threshold cycle (CT) values. Fold comparisons were made between the elongated and tubular day 16 embryos.

#### Statistical Analyses to Determine Differences in mRNA Expression

Mean mRNA CT values were transformed into linear 2^−ΔCT^ values and differences in relative expressions between elongated and tubular embryos were calculated with a Student’s *t*-test using SAS Analytics software (9.4 version; SAS Institute, Cary, NC, USA).

### 2.5. Protein Immunoblots

Proteins, such as PPARG, RXRG, SLC2A1, SLC27A6, CXCL10, ISG15, DNMT1, ZEB1, HIF1A, and GAPDH (endogenous control) were selected to recognize the presence of protein in day 16 embryos.

Western blots were performed by methods described previously [10,26]. Briefly, aliquots of elongated and tubular conceptus were pooled. An amount of 100 µL of 100% ethanol was added to the organic phase of the pooled elongated and tubular embryo samples (from the RNA isolation step). After thorough mixing and incubation, the sample was centrifuged at 2000× *g* at 4 °C for 5 min and the supernatant was further used. An amount of 0.5 mL of isopropanol was added to the supernatant and incubated for 10 min. Then, the sample was centrifuged at 12,000× *g* at 4 °C for 10 min and the proteins were pelleted. After washing, the protein was resuspended. Further, electrophoresing proteins (60 μg) through 12% SDS-PAGE gel (Bio-Rad Laboratories, Philadelphia, PA, USA), transferring onto polyvinylidene difluoride (PVDF) membrane (Bio-Rad Laboratories), blocking non-specific binding, incubating at 4 °C overnight with primary antibodies (Appendix A), washing in wash buffer containing 2% animal serum and 0.1% detergent, and incubating in secondary antibodies (Appendix A) conjugated with FITC fluorophore were sequentially performed. Following 1 h incubation with secondary antibodies at room temperature, the membranes were washed. The membranes were scanned using the Pharos FX Plus system (Bio-Rad Laboratories). Conjugated FITC fluorophore was excited at 488 nm and read at the emission wavelength of 530 nm.

## 3. Results

### 3.1. Identification of Differentially Expressed miRNAs in Day 16 Embryo

Fold expression of 84 prioritized miRNAs in day 16 embryo between fertile and repeat breeder cows were presented in Appendix A. The day 16 embryo miRNA analysis revealed 19 miRNAs (11 upregulated and 8 down regulated; ≥2-fold expression; *p* < 0.05) were differentially expressed in elongated embryo from fertile, normal cows compared with tubular embryos from subfertile repeat breeder cows (Figure 1).

### 3.2. In Silico Analysis

Nucleotide sequence similarities for the 19 differentially expressed bovine miRNAs and corresponding miRNAs for humans are presented in Appendix A. Bovine sequences were very similar to human nucleotide sequences. Therefore, human miRNA IDs were used to construct the miRNA-mRNA interaction network and functional enrichment analysis.

The miRNA-mRNA interaction analysis for 19 DE-miRNAs revealed associations of a total of 66 miRNAs and 201 target genes (Appendix A) with 12,559 nodes and 39,159 edges (Figure 2; miRNet, http://www.mirnet.ca/ accessed on 10 July 2023).

The protein–protein-interaction analysis (STRING, accessed on 11 July 2023) of the target genes (PPI enrichment *p* < 1.0 × 10^−16^) found 1154 significantly enriched gene ontology (GO) biological processes terms (false recovery rate, *p ≤* 0.05) and 107 significant (false recovery rate, *p ≤* 0.05) KEGG enrichment pathway terms (Appendix A). Among the enriched biological processes, 38 processes including regulation of cellular process, metabolic process, and anatomical structure development had more than 100 background genes (False Discovery Rate *p ≤* 0.05). Interestingly, embryo development and placenta development gene ontology (GO) terms were regulated by 33 and 5 background genes, respectively (Table 1).

The constructed PPI network for predicted genes of the DE miRNAs is given in Figure 3. Further, the top 20 hub genes (MCC method) are presented in Figure 4. Identified hub genes and GO annotation terms are presented in Appendix A.

Differentially expressed miRNAs predicted gene list and its linked GO terms, such as embryonic placenta (preimplantation) formation, implantation, and placenta (post implantation) formation process are shown in Appendix A. Interestingly, there were 118 genes that regulated 20 GO terms of the embryonic placental development, 77 genes were identified in regulating 5 GO terms of implantation, and 184 genes participated in regulating 25 GO terms of the placental development. Figure 5, a Venn diagram, shows the number of total and common genes participating in these three developmental processes. Interestingly, four genes, leukemia inhibitory factor (LIF; Gene ID: 280840), nodal (NODAL; Gene ID: 530748), synapse defective Rho GTPase homolog 1 (SYDE1; Gene ID: 789472), and tripartite motif containing 28 (TRIM28; Gene ID: 519422), were observed to be common to these three developmental processes.

### 3.3. mRNA and Protein Expressions of Selected Candidates

The mRNA expressions of PPARA, PPARD, PPARG, RXRA, SLC2A1, SLC27A6 CXCL10, DNMT1, ZEB1, HIF1A, and ISG were greater (*p* < 0.05) in the elongated conceptus compared to the tubular conceptus (Figure 6); whereas, the mRNA expressions of RXRB and RSAD2 were similar in the elongated conceptus compared to the tubular conceptus (*p* > 0.1).

Protein samples were electrophoresed and the presence of PPARG (57 KDa), RXRG (51 Kda), SLC2A1 (54 Kda), SLC27A6 (70 Kda), CXCL10 (9), ISG15 (18 Kda), DNMT1 (185 kDa), ZEB1 (124 kDa), HIF1A (93 kDa), and GADPH (36 Kda) proteins (Appendix A) was recognized.

## 4. Discussion

In this study, 19 DE-miRNAs were identified in day 16 preimplantation embryos. In silico analysis of miRNAs and their associated mRNAs revealed their crucial participation in embryo development, development of placenta, establishment of pregnancy, and fetal organ development. Several hundred gene transcripts were found in gametes and early embryo stages, indicating critical roles of these genes in bovine embryo development, which is crucial for pregnancy establishment. It is plausible that these are transcriptionally inactive in mature gametes and early embryos. Previous studies identified several genes expressed exclusively in specific embryo stages during preimplantation [35,36,37]. The stage-exclusive transient expression suggests that these genes are critical only for specific phases of development. The target genes predicted GO terms pertinent to embryo and placental development observed in the current study were corroborating their role in these developmental processes during early pregnancy establishment. A previous study identified proteins produced by the day 16 bovine conceptus from the uterine luminal fluid [38]. The target genes predicted by the DE-miRNAs in day 16 conceptus in the current study were identical to the target genes associated with the proteins isolated in this previous study.

Gene clusters, groups of target genes partaking in a particular function, display higher expression in competent elongated embryos than in noncompetent tubular embryos, which translates the developmental capacity of bovine embryos [39]. Gene clusters mainly associated with mitochondrial functions include ATP synthases, eukaryotic translation initiation factors, ribosomal proteins, mitochondrial ribosomal proteins, NADH dehydrogenases, cytochrome c oxidases, aldehyde dehydrogenases, proteasomes, WD repeats, and keratins (Figure 7). Greater abundances of these gene clusters in the competent embryos indicate the upregulation of global protein translation turnover and ATP generating pathways.

Differentially expressed miRNAs predicted target gene clusters were mainly associated with gene ontology annotations regulating embryo, placenta, and in utero embryonic development (Figure 8). These clusters included the genes *ACVR1B, ADD1, ARHGAP35, BMPR2, BRCA2, CDX2, COL1A1, DAD1, DNMT1, DNMT3A, FOXC1, GATA6, GNAS, ITGA5, HES1, HIF1A, KDM6B, KMT2A, MAFG, NOTCH1, NOTCH2, PRKACB, ROCK2, SATB2, SKI, SMAD3, SMAD5, SOX4, ST14, TGFBR2, ZEB1, ZEB2,* and *ZFP36L1*. The number of genes involved in each biological function associated with embryo, placenta, and in utero fetal development ranged from 3 to 33, displaying the importance of the interaction in regulating these functions (Figure 9).

On in silico analysis in the current study, 27 pathways were involved in the embryo, placenta, and in utero fetal development (Figure 10). Interestingly, chemokine signaling influences pathway that is important for regulating embryo attachment and placentation, leading to embryo survival [40,41,42]. Genes involved in chemokine signaling were expressed more in the inner cell mass than in the trophectoderm of the bovine embryo and participate in spatial organization of the inner cellular mass [43]. In ewes, chemokine ligand twelve (*CXCL12*) expression was greater in the trophoblast on day 22 and 24 and in naturally mated ewes compared to ewes that received embryos [44]. Increased *CXCL12* expression promotes implantation and placentation. Decreased *CXCL12* in the IVA embryos may compromise the pregnancy establishment. Further, Forkhead box O (*FOXO*) transcription factors are crucial regulators of many genes that control a wide range of cellular functions, including differentiation, homeostasis, and survival. Mice lacking *FOXO3* manifest age-related female infertility [43,44], whereas targeted disruption of *FOXO1* leads to midgestational death, attributed to abnormal vascular/cardiovascular morphogenesis [45,46,47]. It should be noted that all pathways are involved in various capacities in regulating early embryogenesis and pregnancy establishment.

The protein-to-protein functional association from the target genes revealed 90 GO terms associated with cellular processes. The top 25 cellular processes associated GO terms with the most observed gene counts and top 25 cellular processes associated GO terms with the lowest false discovery rates are shown below (Figure 11). Cellular process (189 genes), cell communication (102 genes), cell differentiation (90 genes), cell motility (28 genes), cell migration, and cell morphogenesis (25 genes) substantiating that the DE-miRNAs and their associated genes played vital role in regulating embryo, placenta, and embryo/fetal organ development. Further, cellular response to stress (50 genes) and apoptosis (46 genes) indicates that though these genes are vital for cellular and development processes, their aberrant expression could lead to abnormal embryonic and/or placental development.

In cattle, early embryonic development from the blastocyst (day 7, ~174 μm) to the elongated conceptus (day 16, ~60 mm) occurs over a span of 16 days [4,5]. The embryo rapidly expands from thousands to millions of cells and develops nearly all major organ systems. Studies profiled the transcriptomes of millions of cells during the early stages of gestation and viewed the whole developmental processes during embryogenesis and organogenesis [48,49]. The dynamics of gene expression within hundreds of cell types and trajectories over time were explored [48,49,50], and fate-mapping and lineage-tracing studies of spatiotemporal transcriptomes showed key cellular processes during embryogenesis similar to processes identified in the current study. Further, single-cell transcriptomes compared cell types across species, identified conserved and divergent transcriptional programs regulating organogenesis and cell fate decisions, and the molecular basis of tissue regeneration during embryonic development [48,49,50]. The target genes predicted by DE-miRNAs in this study were also seen in gametes and various stages of early embryos supporting their involvement in spatiotemporal progressions and the regulatory programs of embryo development [51].

There were 18 gene ontology terms associated with “placenta development” including angiogenesis (sprouting of new blood vessels from existing ones) and vasculogenesis (formation of new blood vessels from angioblasts and endothelial cells) linked to the target genes. These are shown with observed gene counts and false discovery rates (Figure 12). Embryonic placenta development (5 genes; Table 2), placenta development (11 genes; *HES1, NOTCH2, NOTCH1, POU4F2, APP, BTG2, CRK, DPYSL2, RERE, HMGB1, ADARB1, PTEN, CXCL12, PAFAH1B1, BCL2, RDX, ARHGAP35, CREB1, RHOA, ROBO1, PIK3R1, DICER1,* and *ZEB2*), vascular development (27 genes), regulation of apoptotic process (60 genes), angiogenesis (42 genes), vasculogenesis (8 genes; *ITGB8*, *NOTCH1*, *FBXW7, SPRED1, CAV1, TGFBR2, QKI,* and *ZFP36L1*), response to oxidative stress (15 genes), and responses to vitamin A (3 genes; *DNMT3A*, *PPARG*, and *DNMT3B*) and vitamin E (3 genes; *COL1A1*, *CCND1*, and *PPARG*) substantiated the DE-miRNAs and their associated genes’ vital roles in regulating placenta development.

There were 59 GO terms associated with “organ development”, including in the embryonic and fetal stages, linked to the target genes (Figure 13). The genes targeted by DE-miRNAs in the current study regulate the development of organs (brain, heart, lung, gastro-intestines, kidney, reproductive tract, and gonads) and systems (circulatory, nervous, musculoskeletal, and lymphatic).

In general, genes associated GO terms, such as embryo development, embryonic placenta development, placenta development, cell differentiation, cell motility, cell migration, cell morphogenesis, and different organ system development were critical processes.

Preimplantation embryos undergo intense resetting of epigenetic information inherited from the gametes. Genome-wide analysis of single-bases found similarities of epigenetic information but there were species differences in DNA methylation patterns and reprogramming of epigenetic function [4]. A pattern of methylation that is seen in oocytes persists throughout the cleavage stages (blastocysts and day 16 embryos), albeit with some methylation level reduction in cows.

The *SMAD* family, including *SMAD1 to 7*, target genes were connected in the current study. *TGFβ* ligand binds to their receptors, triggering *SMAD5* (*BMP*s) and *SMAD3* (*TGFβ*, activin and nodal) downstream pathways [52]. *SMAD1/5* signaling is required for blastocyst production, first cell lineage determination as well as mRNA and protein regulation of *CDX2* cell markers [53]. *SMAD1/5* signaling was essential for embryotropic functions during days 4–7 but not days 1–3 of embryo development, suggesting a temporal role in bovine embryos [53]. *SMAD3* inhibitor (SIS3) administration showed a reduction in insulin-like growth factor-1 (*IGFBP-1*) expression in the uterus and consequently decreased the number of implanted embryos, demonstrating that *TGF-β/SMAD3* signaling is involved in the process of embryo implantation [54]. *SMAD5* is expressed in the extra-embryonic and embryonic regions. SMAD transcription factors are vital for maintaining the structural and functional integrity of the uterus, which is required for the establishment and maintenance of pregnancy [55]. Uteri from the SMAD1/5/4-Amhr2-cre KO mouse model exhibited multiple defects and led to a hostile uterine lumen which diminished embryo implantation [55]. The defective uterine decidualization led to pregnancy loss. Amniogenesis was abnormal in the embryos deficient in the bone morphogenetic protein (BMP) signaling effector SMAD5, displaying delayed closure of the proamniotic canal, aberrant amnion, and folding morphogenesis [56]. Mice lacking *SMAD 5* exhibited angiogenesis defects and mesenchymal apoptosis [57]. These findings supported the findings noted in the current study that target genes *SMAD3* and *SMAD5* are involved in embryo development, embryo morphogenesis, embryo organ development, and in utero fetal development processes in cows.

In the current study, *DNMT* family (*DNMT1*, *DNMT3A* and *DNMT3B)* was recognized as regulating many biological processes including DNA methylation involved in embryo development (GO:0009790), anatomical structure development (GO:0048856), the cellular process (GO:0009987), positive regulation of cell population proliferation (GO:0008284), the apoptotic process (GO:0006915), positive regulation of the developmental process (GO:0051094), positive regulation of the metabolic process (GO:0009893), and system development (GO:0048731). The DE-miRNAs linked to the *DNMT* transcripts in the current study are given in Table 2. Regarding the *DNMT* family (*DNMT1, DNMT3A, DNMT3B*), transcripts of the de novo methyltransferases *DNMT3A* and *DNMT3B* as well as the maintenance methyltransferase *DNMT1* and its supplementary protein UHRF1, were readily detected in pig and cow oocytes [4]. Despite the considerable global methylation levels until the morula in both pigs and cows, *DNMT1* and *UHRF1* transcript levels decline markedly. In cows, *DNMT3A* and *DNMT3B* transcript levels recovered strongly from the morula to the blastocyst stage, which could be associated with the onset of de novo methylation observed in the blastocysts [4,58]. In cows, DNMT1 was found in the cytoplasm in zygotes but not in the 8–16 cell embryos where it had a mostly nuclear location [59]. The DNMT1 in cow at 8–16 cell stages could be important before de novo methylation by DNMT3A and DNMT3B at the blastocyst stage. DNMT3A and DNMT3B significantly increased in expression from the morula to the blastocyst stage in cows [4]. DNMT1 protein was important in maintaining methylation in embryos undergoing cleavage (short maternal DNMT1 protein), and therefore, the longer DNMT1 isoform was important during the post-implantation stage [60]. *DNMT1* expression was reduced in early placentae from sheep IVP embryos, consequently resulting in growth arrest and the death of embryos. In contrast, normal levels of *DNMT1* and its cofactors were observed in placentae from the IVP embryos that survived. The DNA methylation system was severely compromised in IVP placentae up to day 24. However, the low DNMT1 enzymatic activity in IVP placentae that persisted after this stage was not lethal for the developing embryos [61]. Abnormal DNA methylation of imprinted genes, placenta-specific genes, immune-related genes, and sperm DNA may affect embryo implantation, growth, and development, directly or indirectly, leading to recurrent pregnancy loss (RPL) [62]. About 4% of spontaneous abortions were linked to abnormal DNA methylation [63]. As previously mentioned, the methylation and expression status of imprinted genes are tissue and are stage specific during development [64,65]. More than 50% of known imprinted genes are expressed in placenta, and they are important to cellular differentiation and embryonic development [58,59]. *DNAMT3A* [66] and the *DNMT3B* [67] gene polymorphism may be a potential genetic markers for RPL risk. Abnormal DNA methylation of imprinted genes can lead to abnormal silencing of active alleles or abnormal expression of inactive alleles, causing imprinting disorders or deletions, fetal neurodevelopmental defects, and metabolic disorders, which in turn affect embryo development and result in poor pregnancy outcomes [62].

The target genes *NOTCH1* and *NOTCH2* were involved in embryonic placenta and in utero fetal development, embryo morphogenesis, and embryo organ development in the current study. On implantation, NOTCH signaling was triggered via cell–cell interaction, in which *NOTCH* receptors expressed on the surface of the endometrium bound to ligands present on the surface of blastocysts [68]. Activin A or *TGFβ* upregulated several transcription factors, such as *SNAI2, ZEB1,* or *TWIST1* mediated by phosphorylated *SMAD2* or *SMAD3* [69,70]. NOTCH signaling upregulates the *SMAD*3 expression, which enhances *TGFβ*-induced epithelial–mesenchymal transition (EMT) marker expression [71,72,73]. These findings suggest that attachment of trophectoderm cells to endometrial epithelial cells could activate NOTCH signaling, which enhances activin A-induced EMT marker expression via SMAD2, SMAD3 and/or SMAD4, which regulates sequential events of implantation [74,75,76].

Further, *ACTL6A* (mir-25a) and *BYSL* (mir-128-3p, mir-128-3p, mir-320a, mir-17-5p, mir-191-5p, mir-200b-3p, hsa-mir-210-3p) involved in blastocyst formation; *ATP1B1* (miR-218-5p)*, CDX2* (miR-181b-5p), *CDK11B* (miR-191-5p), *DMPT1, TGFBR1* (miR_199a-5p), *TGFBR3, and XAB2* involved in regulation of blastocyst development; *ANKRD7, SMARCA4, and SMARCB1* were linked to blastocyst hatching; *BYSL, CDH1*, SP1, and SP3 were associated with trophectoderm cell differentiation; *CHEK1*, *COPS2*, *DMPT1*, and *TET1* were related to inner cell mass proliferation, regulating diverse processes of pregnancy establishment.

mRNA expressions of *PPARG, RXRG, SLC2A1, SLC27A6, CXCL10,* and *ISG15,* were greater in elongated, competent day 16 embryos from normal cows compared to the tubular, noncompetent day 16 embryos. This finding was consistent with our previous study [8]. In addition, *DNMT1, ZEB1*, and *HIF1A* mRNA expressions were also greater in elongated, competent day 16 embryos from normal cows compared to the tubular, noncompetent day 16 embryos. It is important that these genes and their downstream signaling cascades play important roles in embryo elongation and interruptions in cross-talk between endometrium and conceptus, impaired conceptus elongation, and pregnancy establishment in repeat breeder cows [10].

The selection of cows representing the normal, fertile, and subfertile repeat breeder groups for this study was reliable [8,77,78]. During selection, care was taken to minimize the impact of other factors influencing fertility. Subfertility due to repeat breeding in lactation is immensely important in a production system. Days open (interval from caving to conception) is a primary, economically important, and dependable reproductive parameter. An increase in the quantity of insemination/pregnancy is positively related to an increase in days open. An increase in days open translates to production loss and economic loss. Further, the probability of subfertile repeat breeder cows being culled (citing poor reproductive performance) is greater compared to normal cows. This supports the notion that the repeat breeder cows were subfertile by definition and performance. It should be noted that those cows continually showed normal and repeat breeder performances accordingly in later lactation as per the records of this study.

Based on the findings from this study, it is evident that miRNAs orchestrated the complex mechanisms of embryo, placental, and organ development by interacting with genes and transcription factors. The limitations of the data set presented in this study are (i) gene ontology information for the investigated set of candidate bovine miRNAs and known developmentally important genes; (ii) included in vivo-derived embryos from fertile and subfertile cows, not involving in vitro or cloned embryos. The discovered miRNA-mRNA interactions in this study were mostly based on predictions from public databases. Thus, the causal regulations of these interactions and mechanisms require further functional characterization.

## 5. Conclusions

In conclusion, miRNAs were differentially expressed in the elongated, competent day 16 embryos from normal cows compared to the tubular, noncompetent day 16 embryos from subfertile cows. The DE-miRNAs in day 16 embryos elucidated in this study plausibly regulated critical pathways that are required for pregnancy establishment and associated genes at the transcriptional or post-transcriptional level. In silico network analysis associated DE-miRNAs and its predicted genes with the tasks of early embryo development, implantation, placental development, and organogenesis.

## Figures and Tables

**Figure 1 animals-13-03052-f001:**
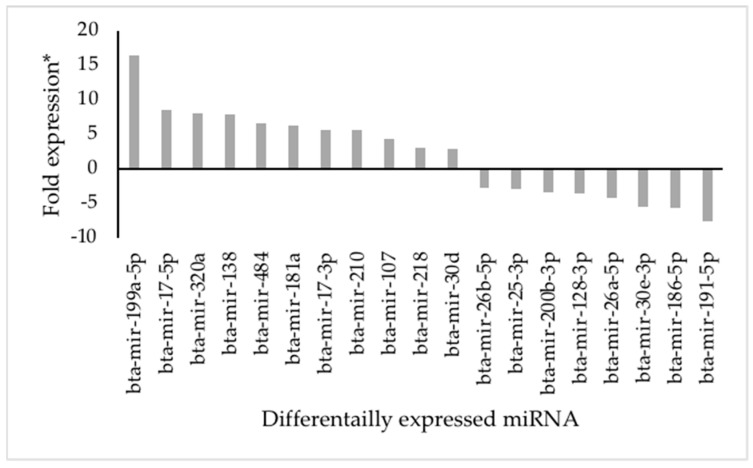
Fold regulation of differentially expressed day 16 embryo miRNAs in the elongated conceptus compared to the tubular conceptus *. A total of 11 upregulated (≥2) miRNAs and 8 down regulated (≤−2) miRNAs (*p* < 0.05) were shown.

**Figure 2 animals-13-03052-f002:**
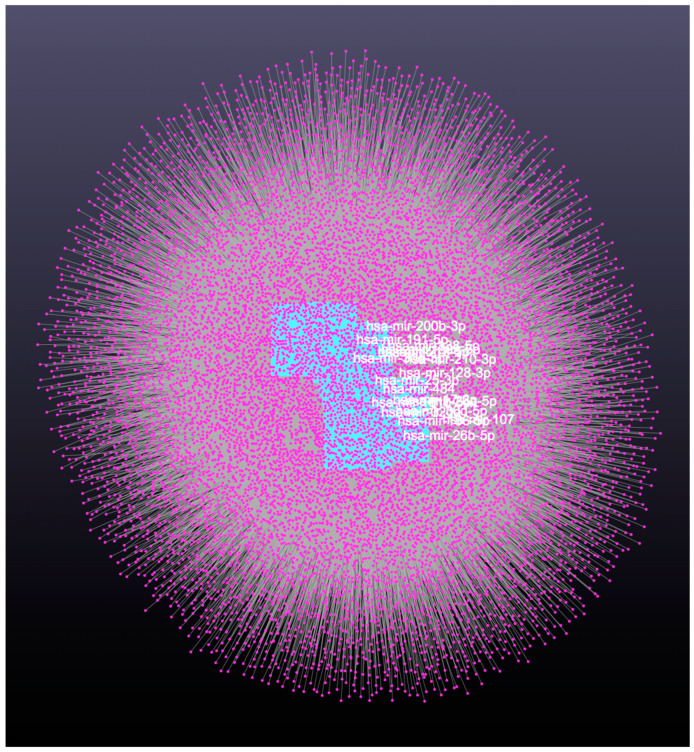
miRNA-mRNA interaction network of day 16 embryos of dairy cows. Blue squares represent miRNA. There were 66 miRNAs, overlapping each other, included in Appendix A. Pink dots represent genes. Grey lines represent interactions between miRNAs and genes.

**Figure 3 animals-13-03052-f003:**
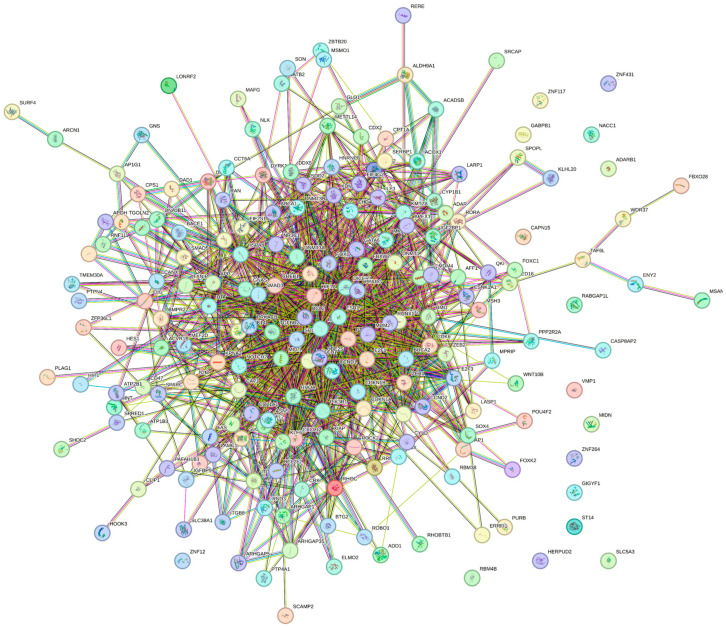
STRING protein–protein interaction (PPI) network. PPI network for the DE-miRNAs predicted 201 target genes (201 nodes and 1078 edges, PPI enrichment *p* < 1.0 × 10^−16^). The color nodes represent proteins. The edges represent interactions.

**Figure 4 animals-13-03052-f004:**
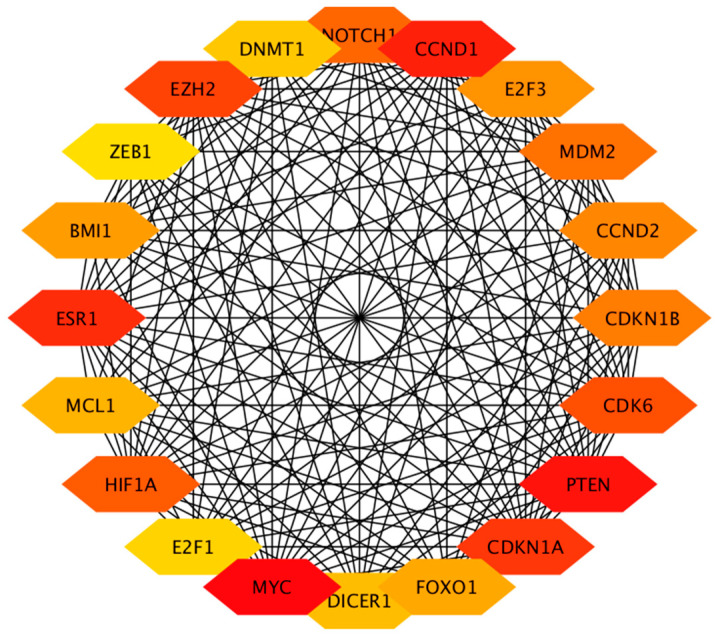
Interaction of hub genes of differentially expressed miRNAs in the protein–protein interaction network. Colors red to yellow denote high to low degrees of expression. Black lines indicate interactions between genes.

**Figure 5 animals-13-03052-f005:**
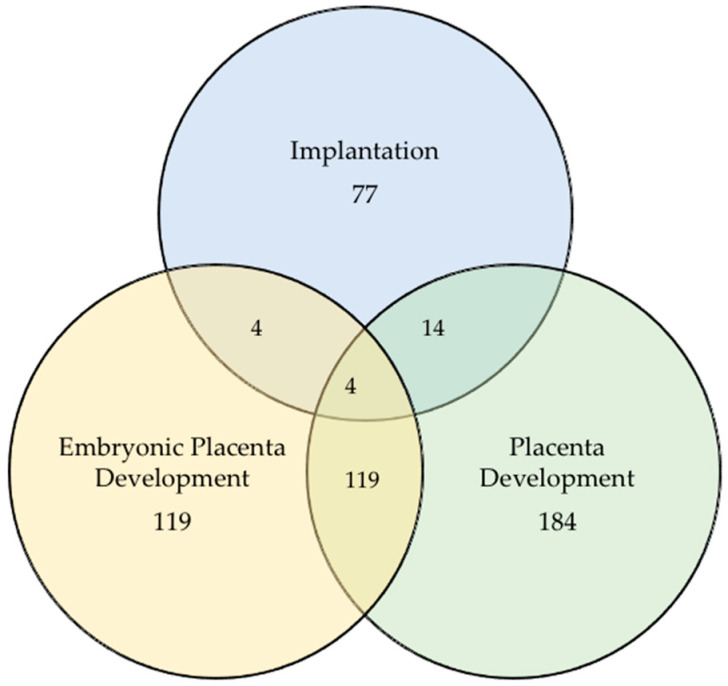
Venn diagram showing number of target genes participating in embryonic placental development, implantation, and placenta development (post-implantation).

**Figure 6 animals-13-03052-f006:**
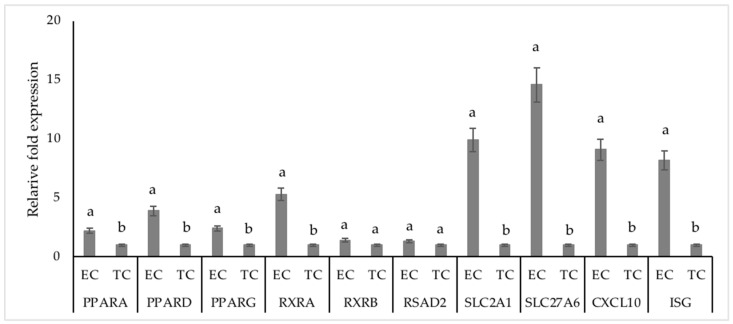
mRNA expression of target genes in elongated vs. tubular day 16 embryos in dairy cows. ab bars with different superscripts between EC and TC within mRNAs were significant (*p* < 0.05); EC, elongated conceptus—≥25 mm long; TC, tubular conceptus—10 to 20 mm long; PPAR—Peroxisome proliferator-activated receptor; RXR—Retinoid X receptor; RSAD2—Radical S-Adenosyl Methionine Domain Containing 2; SLC2A1—Solute Carrier Family 2 Member 1; SLC27A6—Solute Carrier Family 27 Member 6; CXCL10—C-X-C motif chemokine ligand 10; ISG15—Interferon interferon-stimulated gene-15; BRP—bovine ribosomal protein (endogenous control).

**Figure 7 animals-13-03052-f007:**
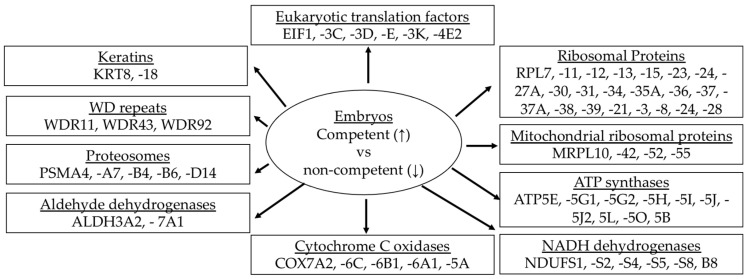
Gene clusters significantly enriched in the competent embryos compared to the noncompetent ones. ↑ increased expression; ↓ decreased expression.

**Figure 8 animals-13-03052-f008:**
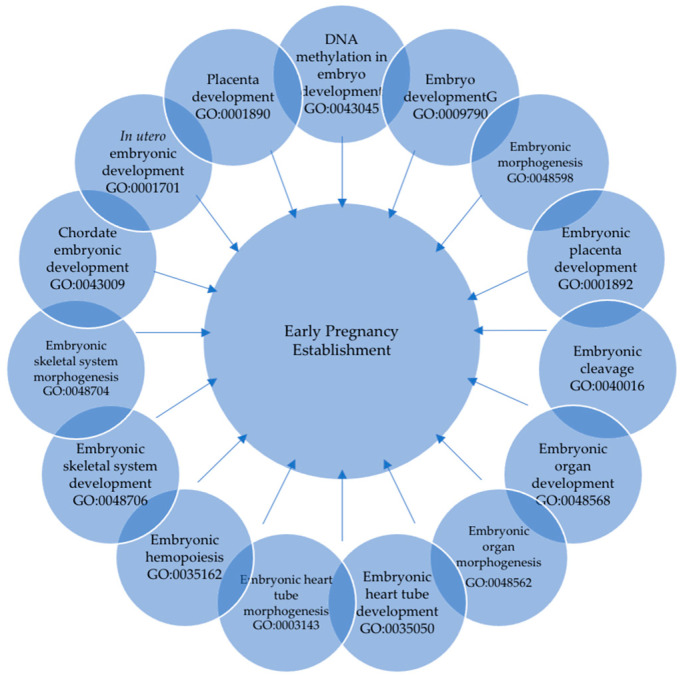
Gene ontology and term descriptions pertinent to “embryo/placenta” predicted by target genes associated with differentially expressed miRNAs in day 16 bovine conceptus.

**Figure 9 animals-13-03052-f009:**
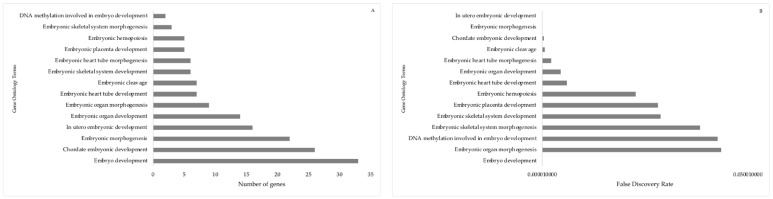
Gene annotation terms specific to embryo developmental functions predicted by target genes are associated with DE-miRNA in day 16 embryos in dairy cattle. (**A**) Number of target genes involved in the gene ontology embryo developmental functions. (**B**) False discovery rate for gene ontology embryo developmental functions. The bar plot shows the *p*-value of the selected GO terms.

**Figure 10 animals-13-03052-f010:**
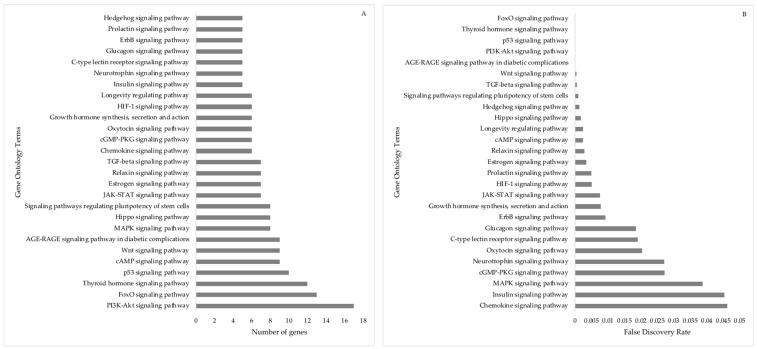
Signaling pathways involved in the embryo developmental functions. (**A**) Number of target genes involved in signaling pathways in the embryo developmental functions. (**B**) False discovery rate for signaling pathways involved in the embryo developmental functions; the bar plot shows the *p*-value of the selected GO terms for STRING biological processes.

**Figure 11 animals-13-03052-f011:**
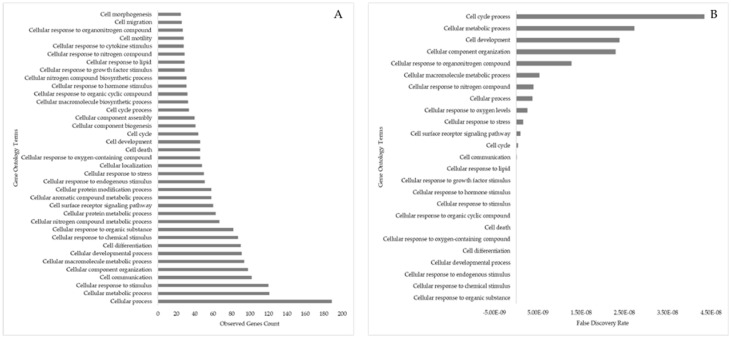
Gene ontology term “cellular process” linked to the target genes (associated with DE-miRNA in day 16 embryos) involved in the embryonic development. (**A**) Number of target genes involved in the cellular process of embryo development. (**B**) False discovery rate for cellular processes involved in embryo development. The bar plot shows the *p*-value of the selected GO terms.

**Figure 12 animals-13-03052-f012:**
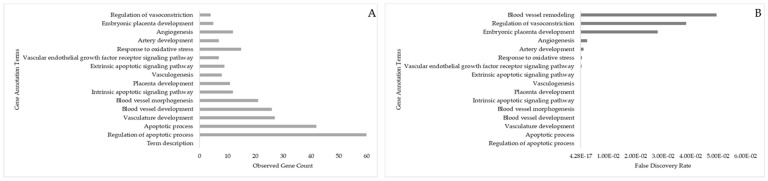
Gene ontology term “placenta development” predicted by target genes (associated with DE-miRNA in day 16 embryos). (**A**) Number of target genes involved in placenta development terms. (**B**) False discovery rate for gene ontology terms involved in placenta development; the bar plot shows the *p*-value of the selected GO terms.

**Figure 13 animals-13-03052-f013:**
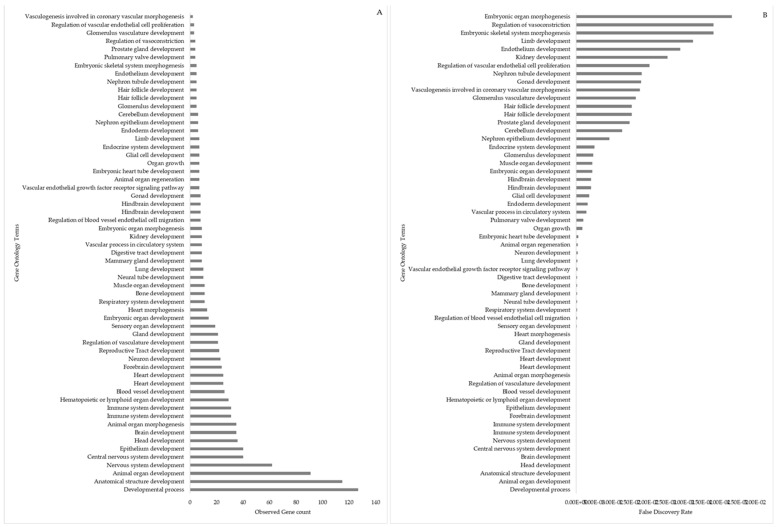
Gene ontology term “embryo/fetal organ development” linked to the target genes (associated with DE-miRNA in day 16 embryos). (**A**) Number of target genes involved in embryo/fetal organ development. (**B**) False discovery rate for gene ontology terms involved in embryo/fetal organ development; the bar plot shows the *p*-value of the selected GO terms.

**Table 1 animals-13-03052-t001:** Gene ontology terms “embryo development” and “embryonic placenta development” biological process with genes and matching proteins.

Gene Ontology Term ID	Term Description	Observed Gene Count	Background Gene Count	Strength	False Discovery Rate	Matching Proteins in Network (Labels)
GO:0009790	Embryo development	33	1002	0.51	2.63 × 10^−7^	COL1A1, HES1, SOX4, DAD1, KDM6B, NOTCH2, DNMT3A, ADD1, GATA6, **NOTCH1** *, ST14, ITGA5, ROCK2, SMAD3, MAFG, TGFBR2, **DNMT1** *, **ZEB1** *, PRKACB, GNAS, BMPR2, SKI, BRCA2, FOXC1, CDX2, ARHGAP35, ZFP36L1, SATB2, KMT2A, **HIF1A** *, SMAD5, ACVR1B, ZEB2
GO:0001892	Embryonic placenta development	5	88	0.74	0.0281	HES1, ST14, CDX2, ZFP36L1, **HIF1A** *

* Hub genes were denoted in bold letters.

**Table 2 animals-13-03052-t002:** Differentially expressed miRNAs in day 16 conceptus linked to the *DNMT* transcripts.

DNMT1	DNMT3A	DNMT3B
bta-mir-17-5p bta-mir-26a-5p bta-mir-218-5p bta-mir-200b-3p bta-mir-17-3p bta-mir-107	bta-mir-200b-3p bta-mir-181b-5p bta-mir-30e-3p	bta-mir-26a-5p bta-mir-200b-3p bta-mir-107 bta-mir-26b-5p bta-mir-320a bta-mir-191-5p bta-mir-210-3p

## Data Availability

The data presented in this study are available in the article or Appendix A herein.

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
