# Peer review of "Differentially Expressed Candidate miRNAs of Day 16 Bovine Embryos on the Regulation of Pregnancy Establishment in Dairy Cows"

_animals, 2023, doi:10.3390/ani13193052_

Round 1

Reviewer 1 Report

The early embryonic elongation is an essential process for the recognition of the pregnancy in cattle. Therefore, specific gene expression and their transcripts involved in this mechanism are crucial for establishing the pregnancy. The study by Kasimanickam VR and Kasimanickam RK focuses on differentially expressed bovine miRNAs from day 16 embryo of fertile and repeat breeder cows and their link to vital functions of embryonic development. Additionally they performed qPCR and Western blot on selected candidates of genes and proteins.

The reviewer appreciates this smart approach and the used method which are very appropriate to study this repeat breeder phenomenon. This are really interesting and valuable findings. However, there are also some limitations in this study. The reviewer is not sure, if the used statistical method is clear and suitable to distinguish between these two groups of embryos with only 4 group mates – especially under the view, that the embryos are grouped according their length. Definitely, this parameter is a crucial one – however no hints regarding the gender and the accurate age (day 16 embryos can differ for many hours, if the time of ovulation is unknown) of the embryos are given. Do they have no influence on the findings? Furthermore, the milk performance differs enormously between the animals – do the authors neglect this parameter or is it included in the statistical analysis?

In general it is an excellent and innovative study using adequate methods however with some weak areas in the statistical approach or in the statistical description.

Some minor comments: Figure 2 and 3 look impressive – however do not generate an added value. 

Author Response

Date 09/19/2023

Dear Editor,

The authors thank both reviewers for their comments and suggestions. We included our response on a point-by-point basis. Please contact me if you have any questions.

Sincerely,

Ram Kasimanickam.

**************************************************************************************

Reviewer 1.

The early embryonic elongation is an essential process for the recognition of the pregnancy in cattle. Therefore, specific gene expression and their transcripts involved in this mechanism are crucial for establishing the pregnancy. The study by Kasimanickam VR and Kasimanickam RK focuses on differentially expressed bovine miRNAs from day 16 embryo of fertile and repeat breeder cows and their link to vital functions of embryonic development. Additionally, they performed qPCR and Western blot on selected candidates of genes and proteins.

The reviewer appreciates this smart approach and the used method which are very appropriate to study this repeat breeder phenomenon. This are really interesting and valuable findings. However, there are also some limitations in this study. The reviewer is not sure, if the used statistical method is clear and suitable to distinguish between these two groups of embryos with only 4 group mates – especially under the view, that the embryos are grouped according their length. Definitely, this parameter is a crucial one – however no hints regarding the gender and the accurate age (day 16 embryos can differ for many hours, if the time of ovulation is unknown) of the embryos are given. Do they have no influence on the findings? Furthermore, the milk performance differs enormously between the animals – do the authors neglect this parameter or is it included in the statistical analysis?

In general, it is an excellent and innovative study using adequate methods however with some weak areas in the statistical approach or in the statistical description.

Authors: We thank the reviewer for the positive comments.

The cohorts are similar in ovulation timing and milk production. The average milk production was 32.3 ± 2.6 and 32.3 ± 2.2 kg of 3.5% fat-corrected milk for normal and repeat breeder groups, respectively. 

The sample size is calculated and was adequate to distinguish expected differences.

These were included in the revised version for clarity.

Some minor comments: Figure 2 and 3 look impressive – however do not generate an added value.

Authors: We appreciate the comments, but figures are included to show the depth/strength of miRNA-gene and protein-protein interactions.

Reviewer 2 Report

The present manuscript presents results from a study of embryos from cows predicted to have a normal pregnancy and those predicted to abort the pregnancy.  Day 16 was chosen because that is the preimplantation period in cows. The study is complete and found 11 genes to be upregulated and 8 to be downregulated in normal embryos compared to embryos that likely would be aborted.  The study was only on 4 embryos in each category, which could limit the applicability of the results because of various environmental, management, and individual genetic influences.

L 39        Delete from further studies

L 99        Lower case phosphate-buffered saline

L 103     Why and how many?

L 105     superscript degree⁰

L 111     Are these from the subset?

L 441     incomplete sentence

Author Response

Date 09/19/2023

Dear Editor,

The authors thank the reviewer for the comments and suggestions. We included our response on a point-by-point basis. Please contact me if you have any questions.

Sincerely,

Ram Kasimanickam.

********************************************************

Reviewer 2.

The present manuscript presents results from a study of embryos from cows predicted to have a normal pregnancy and those predicted to abort the pregnancy.  Day 16 was chosen because that is the preimplantation period in cows. The study is complete and found 11 genes to be upregulated and 8 to be downregulated in normal embryos compared to embryos that likely would be aborted.  The study was only on 4 embryos in each category, which could limit the applicability of the results because of various environmental, management, and individual genetic influences.

Authors: The sample size was calculated and was adequate to distinguish expected differences. This is included in the revised version.

L 39        Delete from further studies

Authors. We did not change this as it deemed appropriate.

L 99        Lower case phosphate-buffered saline

Authors. Changed as suggested

L 103     Why and how many?

Not sure what this question is pertinent to. Controlling the flush is to obtain intact embryos.

L 105     superscript degree⁰

Authors. Changed as suggested

L 111     Are these from the subset?

 Authors. Yes, please refer to Lines 109 to 111.

L 441     incomplete sentence

Authors. The sentence was rephrased.

Reviewer 3 Report

Please see attached file "Review" 

Author Response

Date 09/19/2023

Dear Editor,

The authors thank the reviewer for the comments and suggestions. We included our response on a point-by-point basis. Please contact me if you have any questions.

Sincerely,

Ram Kasimanickam.

**********************************************************************************

Reviewer 3.

Comments on the paper “Differentially expressed candidate miRNAs of day 16 bovine embryo on the regulation of pregnancy establishment in dairy cows” This paper aims to compare the expression of selected miRNA fragments in embryos derived from cows with a history of repeat breeder with those from cows with a history of normal fertility, to describe their networking, and to predict their biological effects on the basis of known/database values. Despite the complexity of this topic, the authors managed to present it quite clearly. I have some individual annotations or comments which I refer to the corresponding line:

Line 50 – ‘6 cm long’: I recommend inserting a reference at this point. Furthermore, it would be more accurate to specify that it is the whole conceptus, and not the embryo proper, that reaches 6 cm at this stage.

Authors: References were included and changes were made as suggested.

Bazer, F. W., Song, G., & Thatcher, W. W. (2012). Roles of conceptus secretory proteins in establishment and maintenance of pregnancy in ruminants. Asian-Australas J Anim Sci 2012, 25, 1-16. https://doi.org/10.5713/ajas.2011.r.08

Brooks, K.; Burns, G.; Spencer, T.E. Conceptus elongation in ruminants: roles of progesterone, prostaglandin, interferon tau and cortisol. J Animal Sci Biotechnol 2014, 5, 53. https://doi.org/10.1186/2049-1891-5-53

Line 88 – “The selected cows”: it would be interesting to know how many cows were flushed to obtain the 4 embyos/conceptuses (or concepti) for each category and, in case were more than 4, why some were excluded.

Authors: This has been discussed in our previous publications and is not scope this current study. References provided below are included.

Kasimanickam RK, Kasimanickam VR. mRNA Expressions of Candidate Genes in Gestational Day 16 Conceptus and Corresponding Endometrium in Repeat Breeder Dairy Cows with Suboptimal Uterine Environment Following Transfer of Different Quality Day 7 Embryos. Animals (Basel). 2021 Apr 11;11(4):1092. doi: 10.3390/ani11041092. PMID: 33920430; PMCID: PMC8070175.

Kasimanickam RK, Kasimanickam VR. IFNT, ISGs, PPARs, RXRs and MUC1 in day 16 embryo and endometrium of repeat-breeder cows, with or without subclinical endometritis. Theriogenology. 2020 Dec;158:39-49. doi: 10.1016/j.theriogenology.2020.09.001. Epub 2020 Sep 3. PMID: 32927199.

Another unclear point is: why was it necessary to select cattle with a history of repeat breeding? The embryos/conceptuses were later categorized under the microscope anyway, on a morphological basis. Is it just a question of probability of finding them, or are there other reasons?

Authors: As we indicated, the selection was to delineate the differences in embryos from fertile (normal cows and embryos) and subfertile (repeat breeding and developmentally impaired embryo) cows.

Line 110 and following – I suggest reviewing this first sentence, since the definition fertile/subfertile is not accurate in my opinion. In fact, both group of cows only had different history of fertility, but at this point 1) we do not know what would have been in the current lactation (since the sampling started very early in the lactation) and 2) they were in fact all fertile since they made it to the third lactation. What happened to the cows in later lactation: were they inseminated again? If so, did the ones with a history of repeat breeding confirmed their history or became pregnant normally? And what about the other ones, with a better fertility history?

Authors: We very much appreciate the comments. We took careful consideration and applied in selection. We like to indicate that fertility is not about how many lactations, but it is about how many inseminations it takes to get a cow pregnant. It is hugely important in a production system. Increase in number of insemination = increase in days open (interval from calving to conception) = production loss = economic loss & culling.

Also, we like to point out that the probability of repeat breeder cows being culled (citing poor reproductive performance) is greater compared to normal cows. That itself shows that they are subfertile by definition and performance. As per the record those cows continually showed normal and repeat breeder performances accordingly in later lactation.

For e.g., if a cow can reach 5th lactation takes > 3 inseminations and become pregnant at 200 DIM in each lactation, the economic loss will be overwhelming for the operation.

For clarity, the discussion was included in Lines 525 – 535.

Line 210 – it should be “differentially”

The suggestion is incorporated in the revised version.

Line 263 – I suggest improving the description of Figure 2, since it is not clear what each of its parts represent

The explanations were included as suggested in the revised version.

Line 288 and following – For the whole paper, but especially for the discussion, I suggest revisiting the distinction between normal cows and subfertile cows. For this topic I refer to my comment on Line 110 and following. Furthermore, and more importantly: according to my understanding, all analyses made are based on the conceptus itself and not on its interactions with the mother (I apologise in advance if I have misinterpreted or misunderstood any passages on this). However, the provenance of embryos from cows with or without a history of repeat breeding is continually mentioned and especially in the discussion. This, in my opinion, is a methodological error: Since the materials and methods section does not mention the number of cows involved in general, we cannot know whether the fact that all non-competent embryos were derived from repeat breeder cows (and all competent embryos were derived from previously more fertile cows) is coincidental or not.

Given also the small number of subjects considered for each category (n=4) I suggest the authors supplement the information on the cows initially involved (see also my comment on Line 88 and following). Alternatively, the information on previous fertility history should be completely eliminated (or reduced to a mere mention) and the comparison should focus on whether the embryos were, at day 16 and on the basis of morphological criteria, competent or not.

authors: Thank you for this comment. Please refer to the explanation provided for line 88 and 110. We reliably pursued the selection of repeat breeder cows. It was clearly mentioned and explained in all our previous publications. This study is comparing d16 embryos from fertile and subfertile cows. The normal and repeat breeding groups supported by the careful selection of cows and the embryos derived from those cows. Our methods strongly and evidently support this and is not an error.